# Systematic Review and Meta-Analysis of the Correlation Coefficients between Nomophobia and Anxiety, Smartphone Addiction, and Insomnia Symptoms

**DOI:** 10.3390/healthcare11142066

**Published:** 2023-07-19

**Authors:** Lateefa Rashed Daraj, Muneera AlGhareeb, Yaser Mansoor Almutawa, Khaled Trabelsi, Haitham Jahrami

**Affiliations:** 1Department of Psychiatry, College of Medicine and Medical Sciences, Arabian Gulf University, Manama 329, Bahrain; latifarashiddarraj12@gmail.com (L.R.D.); muneeraalghareeb0@gmail.com (M.A.); yaseralmutawa@outlook.com (Y.M.A.); 2High Institute of Sport and Physical Education of Sfax, University of Sfax, Sfax 3000, Tunisia; trabelsikhaled@gmail.com; 3Research Laboratory: Education, Motricity, Sport and Health, EM2S, LR19JS01, University of Sfax, Sfax 3000, Tunisia; 4Government Hospitals, Manama 329, Bahrain

**Keywords:** nomophobia, fear, internet addiction, sleeplessness, systematic review, meta-analysis

## Abstract

Nomophobia is an emerging phenomenon in the 21st century. Consequently, it results in various health problems, both physical and psychological. The following systematic review and meta-analysis aimed to establish the relationship between nomophobia with anxiety, smartphone addiction, and insomnia. To identify the relevant studies, we searched through several databases. Out of the 1523 studies identified, 16 studies met the inclusion criteria. After conducting the statistical analysis, the results revealed that anxiety *r* = 0.31 (95% CI: 0.25 to 0.38), smartphone addiction *r* = 0.39 (95% CI: 0.04 to 0.75), and insomnia *r* = 0.56 (95% CI: 0.38 to 0.75) are positively associated with nomophobia. Mobile phone usage has become inevitable, even for individuals who use it to a lesser degree than others, to perform simple tasks, such as communicating with others or for educational purposes. It is crucial to raise awareness about the consequences of overusing these devices, including the physical and psychological complications in both the short and long terms.

## 1. Introduction

Data and contact technologies have become a crucial part of the generation that we live in. People of all ages have access to technological devices, mainly young people, including high school and college students, who use them to play digital games, study, look up information on the internet, and communicate with their peers [1]. This solidifies that mobile devices are not only used for entertainment and social purposes but extend beyond that to serve as an educational platform used at work and school. However, with the escalating use of smartphones come several health-related problems, such as headaches, repetitive strain injuries, decreased attention, poor concentration, muscle tension, depression, sleep problems, anxiety, and body weight changes. With the extensive use of mobile devices nowadays, an emerging condition called nomophobia is arising [2]. This condition is a disorder of the 21st century, occurring because of technological advancements [3]. Although it is recognized as a type of psychological disorder [3], nomophobia is still not listed in the authorized guidelines of psychiatric diagnoses [4].

Nomophobia, also known as “no-mobile-phobia,” is by definition the anxiety and fear experienced by a person when unable to reach or use their mobile device [5]. Individuals have their phones switched on throughout the day and might even take it to bed with them, due to the persistent fear that they will not have access to the messages that will keep them updated about the recent events and experiences delivered through social media [1]. Those with nomophobia prefer communicating through mobile phones rather than face-to-face contact [4]. 

Researchers have created a number of measures and questionnaires that can be used to quantify mobile phone phobia symptoms [6,7]. These measures often measure how anxious or uncomfortable a person feels when away from their phone, as well as how frequently and intensely they use their phone. The Nomophobia Questionnaire (NMP-Q), which has 20 items and measures many facets of nomophobia, including the fear of losing connectivity, the fear of being unable to speak, and the worry of losing one’s phone, is one regularly used scale [6]. The total score can range from 20 to 140, with higher scores indicating greater levels of nomophobia. Some researchers have suggested using a cutoff score to identify individuals who may be at risk for problematic phone use or nomophobia, but there is no widely agreed upon cutoff score [6]. Participants rank each statement according to how much they agree or disagree with it using a Likert scale [6]. The NMP-Q has a high level of internal consistency and structural validity fitness [6]. 

In the general adult population (aged 18 years and older), nomophobia is becoming more prevalent, with about 20% displaying mild nomophobia symptoms, 50% demonstrating moderate symptoms, and 20% representing severe symptoms [3]. With the upsurge in the occurrence rate arises the issue of smartphone addiction, insomnia, and anxiety. 

The DSM-5 defines anxiety as the excessive worry and apprehensive expectation towards an event or activity lasting at least six months [8]. The anxiety must be excessive and persistent, impacts daily activity, and should not be related to substance abuse or a medical cause. Multiple researches have reported that those who use their mobile phones excessively are more prone to experiencing anxiety episodes when situated at sites with reduced network connection or whenever their phones are out of charge or credits [4]. 

A person is said to have a smartphone addiction when experiencing an uncontrolled and excessive desire to use the phone when it is out of reach. There are a few studies that shed light on the correlation between nomophobia and smartphone addiction. Regardless, it has been established that nomophobia and smartphone addiction are two related terms in that they share common symptoms but differ in their behaviors towards smartphone use. A positive association between nomophobia and social media addiction was addressed in a study by Yildiz Durak, 2019 in Turkish adolescents. Furthermore, it has been suggested that nomophobia might contribute to smartphone addiction, due to the continuous wish to reach the phone. The opposite is true, where social media addiction can potentially lead to nomophobia because of the constant feeling of anxiousness or fear whenever the smartphone is inaccessible [9].

According to the DSM-5, insomnia is considered a sleep–wake disorder characterized by trouble starting sleep, incapability of preserving sleep, or early morning wakefulness lasting for a minimum of three nights per week for a period of at least three months. The condition should not be due to substance abuse or an underlying mental or medical condition that explains insomnia [10]. Individuals suffering from nomophobia face difficulties sleeping without their mobile phones. A Japanese study looking into the sleeping pattern of mobile phone users after lights are out concluded that it is associated with reduced quality of sleep and lower sleeping hours, potentially leading to insomnia [4].

Several studies investigated the relationship between nomophobia and at least one of the three outcomes, which were smartphone addiction, insomnia, or anxiety, individually, rather than all three outcomes [2,3,6,7,11,12,13]. Therefore, the systematic review and meta-analysis below aimed to establish the relationship between nomophobia, smartphone addiction, insomnia, and anxiety. 

## 2. Materials and Methods

### 2.1. Search Strategy

The following meta-analysis and systemic review followed the Preferred Reporting Items for Systematic Reviews and Meta-Analyses (PRISMA 2020) standards. Appendix A provides the compliance checklist. 

A thorough search into the electronic databases was performed, including PubMed (including MEDLINE), PsychINFO, Psychiatry Online, Web of Science, AccessMedicine, Health and Wellness (GALE), ScienceDirect, Cochrane Library (via Ovid), BIOSIS Citation Index, CINAHL, ClinicalKey, EMBASE, PROQUEST Research Library (including ABI/INFORM), Web of Science, and Scopus. The key terms used in the search strategy included “nomophobia” OR “no-mobile-phobia” AND “anxiety” OR “smartphone addiction” OR “internet addiction” OR “digital addiction” OR “social media addiction” OR “insomnia” OR “sleeplessness.” The search involved the published studies between the release date of each database and 15 September 2022. Furthermore, the identified studies’ reference lists were reviewed to find extra relevant studies. 

### 2.2. Inclusion and Exclusion Criteria

The inclusion criteria were (1) nomophobia papers published before the 15 September 2022, (2) original, peer-reviewed studies published in English, (3) articles that assessed the correlation between nomophobia symptoms and at least one of the outcomes of interest (anxiety, smartphone addiction, and insomnia), and (4) participants older than 12 years of age. 

The exclusion criteria were (1) studies which were review articles, case reports, conference abstracts, or book chapters, (2) researches not directed towards the correlation between nomophobia symptoms and at least one of the outcomes of interest (anxiety, smartphone addiction, and insomnia), (3) researches that did not include the relevant information required for the successful completion of data analysis, despite reaching out to the authors, and (4) studies with a sample size fewer than 30 participants. 

### 2.3. Data Extraction and Quality Assessment

The eligible studies were provided to two independent reviews to extract the study design, sample characteristics, measures of nomophobia, anxiety, smartphone addiction, insomnia, and correlation coefficients. Disagreements related to what to include or exclude were resolved with a third reviewer. In cases where relevant information was lacking from any of the studies, the article’s author was contacted through email. We evaluated the quality of the studies, including by using the Newcastle–Ottawa Quality Assessment Scale for cross-sectional studies [14]. 

### 2.4. Measures 

Thirteen measures were utilized in the following study. These measures were classified into three groups: anxiety, smartphone addiction, and insomnia. 

#### 2.4.1. Anxiety Outcomes

Seven measures were used to assess anxiety outcomes, which included the Generalized Anxiety Disorder 7-item (GAD-7), which evaluates general anxiety disorder through a seven-item scale, each of which is assessed on a 4-point Likert scale varying from 0 (never) to 3 (almost every day). The overall score is from 0 to 21 and is classified into mild anxiety (5–9), moderate anxiety (10–14), and severe anxiety (>15) [15]. 

Another measure used was the Social Appearance Anxiety Scale (SAAS), a self-reported questionnaire established by Hart et al. It is a 5-point Likert-type scale that contains 16 items [16]. 

A third measure was the Hospital Anxiety and Depression Scale (HADS), a self-answered questionnaire developed by Nilchaikovit and colleagues. It contains 14 questions divided into 7 odd-numbered questions linked to anxiety and 7 even-numbered questions allied to depression. It is a Likert scale, where each answer is given a score ranging between 0 to 3, whereas the points in each section range from 0 to 21. The cut point of the total score is as follows: 0–7, no psychiatric disorders; 8–10, high anxiety or depression; and 11–21, anxiety or depression [17]. 

Additionally, the Hamilton Anxiety Scale (HAM-A) was used. It is a fourteen-item scale. High scores point towards higher levels of anxiety [18]. The Social Interaction and Anxiety Scale (SIAS) was also utilized. The score on this scale ranges from 0–80, where 0–33 indicates normal, 34–42 indicates probable social phobia, and 43–80 indicates probable social anxiety [19]. 

Furthermore, the Depression, Anxiety, and Stress Scale (DASS) was utilized. The scale comprises of three subscales to assess depression, anxiety, and stress. Each subscale contains 14 statements, and the answers are given on a Likert-type scale from 0 (“did not apply to me at all”) to 3 (“totally or most of the time applied to me”) [20]. Furthermore, the State-Trait Anxiety Inventory (STAI) scale includes 20 items on a 5-point Likert scale [21]. 

#### 2.4.2. Smartphone Addiction Outcomes 

Four measures were used to assess smartphone addiction outcomes, including the Internet Addiction Scale (IAS), a thirty-five-item scale, where each item is given a point from 0 (never) to 4 (very often). The IAS utilizes four symptoms to establish a diagnosis of internet addiction, which are: withdrawal, control problems, functional impairment, and social isolation [22]. 

Furthermore, the Smartphone Addiction Scale—Short Version (SAS-SV) was also utilized. It contains 10 items and is considered a 6-point Likert scale that ranges from 1 (strongly disagree) to 6 (strongly agree) points [23]. 

The Digital Addiction Scale (DAS) was one of the scales used. It is a 5-point Likert-type scale. It includes 29 items divided into three factors: factor 1, “game” (items 1–11); factor 2, “social media” (items 12–23); and factor 3, “impact on daily life” (items 24–29) [5]. The scale of social media addiction among adults was also one of the measures used [24]. 

#### 2.4.3. Insomnia Outcomes

Two measures were used to assess insomnia outcomes. Most of the studies included used the Insomnia Severity Index (ISI), which compromises seven items, with a 5-point Likert scale for each item. The overall score varies from 0 to 28 and is classified as follows: 0–7 (absence of clinically relevant insomnia), 8–14 (threshold insomnia), 15–21 (clinically moderately serious insomnia), and 21–28 (clinically severe insomnia) [25]. The other scale used was the Lebanese Insomnia Scale (LIS-18), which is an 18-item questionnaire formulated in Lebanon. High scores point towards higher levels of insomnia [26]. 

### 2.5. Statistical Analysis

The statistical method for analyzing the relationship between two continuous variables over numerous studies is meta-correlation, also known as meta-analysis of correlation coefficient *r* [27]. Each study’s *r* between the two variables must be calculated, and these correlation coefficients must then be combined using meta-analytic techniques to obtain a general approximation of the strength of association [27]. 

To obtain the standardized effect sizes for each correlation coefficient, we applied Fisher’s r-to-z transformation [27]. By transforming the correlation coefficients into a normal distribution, the overall effect size and standard error could be estimated with greater accuracy [27]. We then estimated the overall correlation between the relevant variables in our meta-analysis using these standardized effect values [27].

R software version 4.1.3 was used to perform statistical analyses on all data. Statistical significance was defined as a *p*-value < 0.05. The ‘metacor’ software was used to carry out the meta-analysis [27]. Results of the meta-correlation analyses were presented visually using the forest plot. 

We assessed heterogeneity mainly using the I-squared (I^2^), H statistic, τ (tau), and τ^2^ (tau-squared) statistics [28,29]. The I^2^ statistic quantified the proportion of total variation in effect sizes that was due to heterogeneity rather than chance [29]. We considered I^2^ values of 25%, 50%, and 75% to represent low, moderate, and high heterogeneity, respectively [29]. The H statistic measured the influence of a single study on the overall results of a meta-analysis [29]. The τ statistic measured the between-study variance in effect sizes and was used to estimate the degree of heterogeneity [28]. A larger τ value indicated greater heterogeneity between studies [28]. The τ^2^ statistic was the estimated variance of true effects across studies after accounting for sampling error [28]. 

Publication bias was evaluated using both the Egger’s test and the rank test [30]. We initially created a funnel plot of the effect sizes vs. their standard errors before running the Egger’s test [30]. The intercept and slope were the key variables of interest in a regression study of the effect sizes on their standard errors [30]. Because small studies with erratic effect sizes were absent from the plot, a statistically significant intercept showed the presence of publication bias [30]. In addition, we applied a modified Egger’s test that took into account the relationship between precision and effect size [30]. 

Assuming no publication bias, we first ranked the effect sizes in ascending order before comparing the observed ranks to the predicted distribution [31]. This was known as the rank test [31]. To calculate the correlation between the observed ranks and their predicted values, we applied the weighted regression method suggested by Peter’s approach [31]. Studies with significant results were more likely to be published and have higher ranks, so a considerable departure from the predicted distribution showed publication bias [31]. 

We also performed other tests, such as the trim-and-fill method and the funnel plot asymmetry test, to further assess publication bias [32]. We interpreted the results of these tests together to gain a more comprehensive understanding of the potential for publication bias in our meta-analysis [32].

Risk-of-bias plots were created for quality assessment using the ‘robvis’ software [33]. The amount of information contained in each judgment was displayed in a summary plot (weighted). The total risk of bias, as well as the bias risk in each area, was shown in a thorough risk of bias assessment of all studies, which was presented using a traffic light plot. 

## 3. Results

### 3.1. Study Identification

We identified 1523 studies during the database search that needed screening. Sixteen studies met the eligibility criteria belonging to this study. Figure 1 shows the PRISMA 2020 study flowchart. The initial inter-rater screening accuracy resulted in a 97% agreement rate between the two reviewers. Nevertheless, after discussion and dialogue with a third senior reviewer (HJ), the agreement rate increased to 100%. This indicated that the third reviewer was able to help resolve any discrepancies or disagreements between the initial reviewers, resulting in a consensus on all of the documents or data that were evaluated. 

### 3.2. Characteristics of the Included Studies 

This meta- analysis included sixteen studies (k = 16, n = 18,209) from nine countries. Study publications ranged from 2018 to 2022. The included studies had sample sizes that ranged from 209 to 5191 participants. The average age of participants ranged from 13 to 33 years. Figure 2 displays a summary plot of the quality assessment, and Figure 3 represents a traffic light plot of the quality assessment of the included studies. Table 1 shows detailed characteristics of the included studies. 

### 3.3. Nomophobia Symptoms with Anxiety (NSA)

A total of k = 8 studies analyzed nomophobia symptoms with anxiety. The analyzed Fisher r-to-z-transformed correlation coefficients varied from 0.21 to 0.45, with most of the values reported to be positive (100%). The approximated Fisher r-to-z-transformed correlation coefficient, depending on the random-effects model, was =0.31 (95% CI: 0.25 to 0.38), as shown in Figure 4. 

Thus, the estimated outcome fluctuated from zero (z = 9.77, *p* < 0.0001). The Q-test revealed a heterogeneity of the outcomes (Q (7) = 72.93, *p* < 0.0001, tau^2^ = 0.01, I^2^ = 88.92%). A 95% prediction interval for the actual outcomes was provided from 0.14 to 0.49. Henceforth, regardless of the presence of some heterogeneity, the actual outcomes of the studies will be directed in the same direction as the approximated average outcome. Table 2 displays a summary of the values. 

An inspection into the studentized residuals showed that values larger than ±2.73 were not found in any of the studies. Therefore, there were no outliers in the content of this model. The Cook’s distances demonstrated that none of the studies had a significant impact. Neither the rank correlation nor the regression test designated any funnel plot asymmetry (*p* = 0.28, and *p* = 0.46, respectively). 

### 3.4. Nomophobia Symptoms with Smartphone Addiction (NSSA) 

Six studies (k = 6) analyzed nomophobia symptoms with smartphone addiction. The analyzed Fisher r-to-z-transformed correlation coefficients varied from −0.05 to 1.15, with most of the values reported to be positive (83%). The approximated Fisher r-to-z-transformed correlation coefficient, depending on the random-effects model, was =0.39 (95% CI: 0.04 to 0.75), as shown in Figure 5. 

**Figure 5 healthcare-11-02066-f005:**
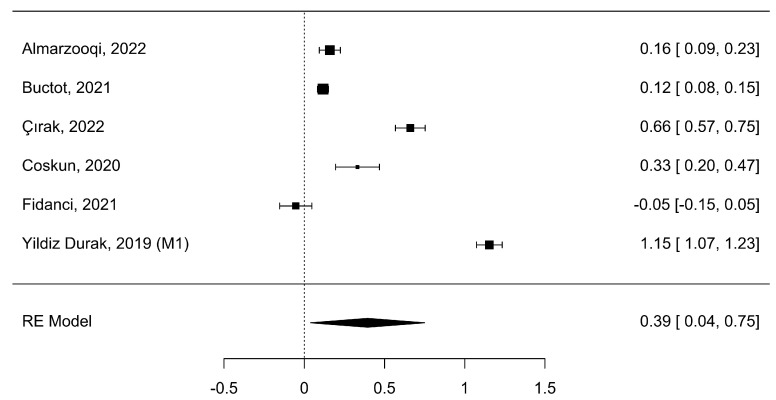
Meta-correlation of nomophobia symptoms with smartphone addiction (NSSA) [5,9,11,35,37,41].

Hence, the approximated outcome varied from zero (z = 2.18, *p* = 0.03). The Q-test revealed a heterogeneity of the outcomes (Q (5) = 675.17, *p* < 0.0001, tau^2^ = 0.20, I^2^ = 99.32%). A 95% prediction interval for the actual outcomes was provided from −0.54 to 1.33. Although the average outcome was appraised to be positive, in some studies, the actual outcome might be negative. Table 2 displays a summary of the values. An inspection of the studentized residuals showed that one study (Yildiz Durak, 2019 (M1), [41]) had a value more significant than ±2.64 and might be an outlier in the content of this model. The Cook’s distances demonstrated that none of the studies had a significant impact. Neither the rank correlation nor the regression test designated any funnel plot asymmetry (*p* = 0.72 and *p* = 0.86, respectively). 

### 3.5. Nomophobia Symptoms with Insomnia (NSI)

Five studies (k = 5) analyzed nomophobia symptoms with insomnia. The analyzed Fisher r-to-z-transformed correlation coefficients varied from 0.26 to 0.74, with most of the values reported to be positive (100%). The approximated Fisher r-to-z-transformed correlation coefficient, depending on the random-effects model, was = 0.56 (95% CI: 0.38 to 0.75), as shown in Figure 6.

**Figure 6 healthcare-11-02066-f006:**
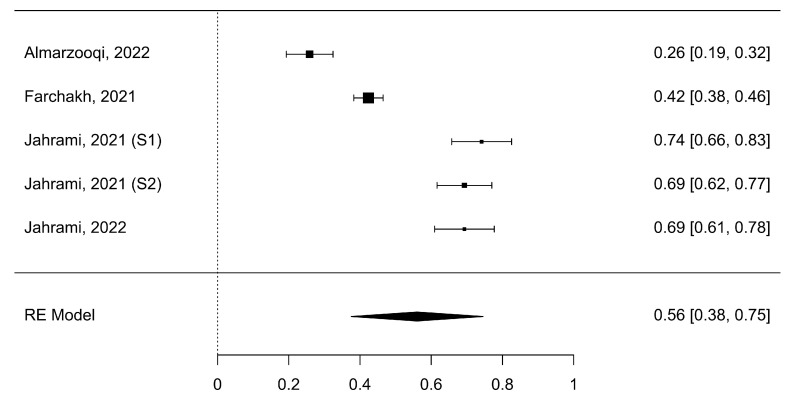
Meta-correlation of nomophobia symptoms with insomnia (NSI) [2,4,11,12].

Consequently, the approximated outcome varied from zero (z = 5.94, *p* < 0.0001). The Q-test revealed a heterogeneity of the outcomes (Q (4) = 141.37, *p* < 0.0001, tau^2^ = 0.04, I^2^ = 97.41%). A 95% prediction interval for the actual outcomes was provided from 0.11 to 1.01. Hereafter, regardless of the presence of some heterogeneity, the actual outcomes of the studies will be directed in the same direction as the approximated average outcome. Table 2 displays a summary of the values. An inspection of the studentized residuals showed that neither of the studies contained values larger than ± 2.58. Therefore, no outliers were detected in the content of this model. The Cook’s distances demonstrated that none of the studies had a significant impact. Neither the rank correlation nor the regression test designated any funnel plot asymmetry (*p* = 0.21 and *p* = 0.20, respectively). 

## 4. Discussion

Studies on the symptoms of nomophobia and their relationships to anxiety, smartphone addiction, and insomnia are becoming increasingly common today. It is possible to draw a conclusion about the relationship between nomophobia symptoms and anxiety, smartphone addiction, and insomnia based on direct evidence from cumulative prevalence data. Based on an in-depth literature review and strict inclusion and exclusion criteria, we identified sufficient studies to conduct a meta-analysis analyzing sixteen studies published from 2018 to 2022. This comprehensive review and meta-analysis revealed that symptoms of nomophobia were positively associated with anxiety, smartphone addiction, and insomnia. 

Anxiety and nomophobia symptoms were evaluated in our study. The results revealed that nomophobia is associated with higher anxiety levels in most individuals. Our findings are in line with published studies [12,42,43]. Individuals tend to develop repeated anxiety attacks whenever they cannot reach out for their mobile phones to access information or use the services it provides. Consequently, the overuse of these devices tends to influence the academic performance of adolescents and, as a result, potentiates anxiety symptoms.

Furthermore, anxiety symptoms are likely to develop and worsen due to social comparison when adolescents compare their lives to social media figures, perceiving their lives as better than theirs [42]. In this line, a person’s mood may be happy or anxious, depending on the interactions they receive from social media [12], not to mention that adolescents diagnosed with an anxiety disorder are more prone to developing a severe form of dependency on the technologies offered by mobile phones, compared to those without anxiety disorder [42]. 

Nomophobia symptoms are associated with higher levels of smartphone addiction. Similarly, Buctot et al. revealed that smartphone addiction and nomophobia are positively intercorrelated in the lifestyle profiles of both senior and junior high school students in the Philippines [9]. Additionally, Çobanoğlu et al. revealed that a positive, moderate correlations were established between nomophobia levels, smartphone addiction, and digital addiction levels [44]. This relationship can be explained by those with low self-regulation and self-control skills using their devices uncontrollably, possessing a higher dependency risk [42]. 

The amount and quality of sleep can be significantly impacted by exposure to blue-light from electronic devices, such as those on laptops, tablets, and cell phones [1,7]. According to one study, people who used electronics for more than four hours before bedtime had more trouble falling asleep and received less restorative sleep than others who did not use them before bed [7]. 

This systematic review and meta-analysis found a substantial correlation between nomophobia symptoms and insomnia. This conclusion is in agreement with those published by other studies [12,13,45]. Specific mechanisms can explain this association. One factor is related to the blue light theory. Blue light emitted from mobile phones lowers the production of the hormone “melatonin”, which promotes sleep and is present in high amounts before bedtime [12]. A systematic review showed that a 2 h contact with shortwave blue light of 400–450 nanometers was adequate to hinder melatonin production. Melatonin production was resumed after refraining from blue lights for about 15 min. Other hormones that are influenced by blue light include cortisol and several markers of the sympathetic nervous system, such as alpha amylase, which promote the persistence of insomnia symptoms. Furthermore, exposure to blue light also seems to activate the prefrontal cortex, predominantly the ventrolateral and dorsolateral areas, which interfere with sleep by increasing alertness and working memory [43]. Secondly, social media is an unlimited process, meaning there is not a clear distinction between where social media interactions begin and end. Hence, many individuals face issues when they resist responding to their messages. They send messages and await a response, the so-called “waiting feature” [13]. Suppose the wait for a response occurs during normal sleep hours. In that case, it will delay their bedtime, leading to a desynchronized sleeping cycle, ultimately affecting sleep quality and leading to insomnia [12]. Thirdly, social media utilization alters an individual’s mood by lowering or elevating it, which may hinder sleep [43]. 

To our knowledge, the review and establishment of the connections between nomophobia, smartphone addiction, insomnia, and anxiety is the first systematic review and meta-analysis. Furthermore, the adequate size of the study contributed to more reliable results. 

However, there are some limitations to this study, including the fact that it focused solely on the targeted variables without accounting for demographic data, such as social status and occupation. Additionally, our study would be enhanced by identifying the risk factors predisposing an individual to developing these symptoms. Finally, moderator analysis was not performed because the limited number of included studies was small; thus, there was less statistical power to find significant changes across subgroups, which could lead to incorrect conclusions. 

## 5. Conclusions

Nomophobia is a rising condition, due to the advancements in technology occurring nowadays. Furthermore, free access to these technologies for people of all ethnicities and ages increases the prevalence rate, reaching up to 20% for those with mild nomophobia, 50% for those with moderate nomophobia, and 20% for those with severe nomophobia; it becomes crucial to raise awareness about the consequences of overusing mobile phone technologies, namely its association with anxiety symptoms, insomnia, and addiction. Measures to assist young adults in managing their use of mobile technologies is vital to promoting adults’ health and well-being as digital technologies become inevitable in our daily lives. 

## Figures and Tables

**Figure 1 healthcare-11-02066-f001:**
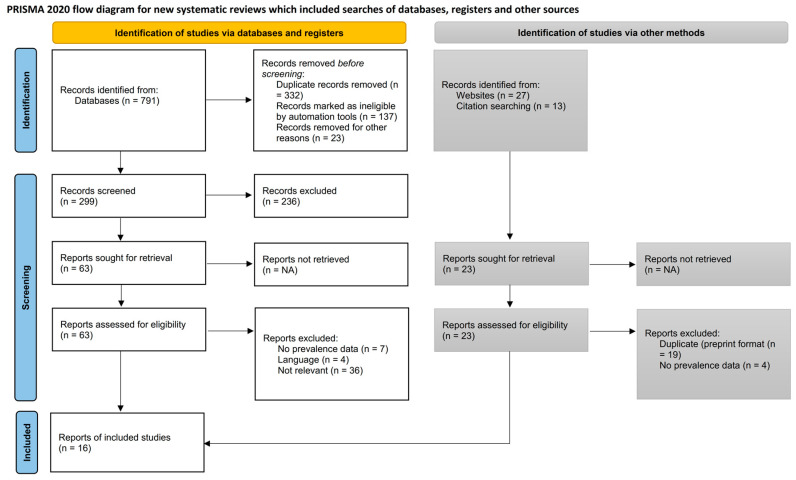
PRISMA 2020 study flowchart.

**Figure 2 healthcare-11-02066-f002:**
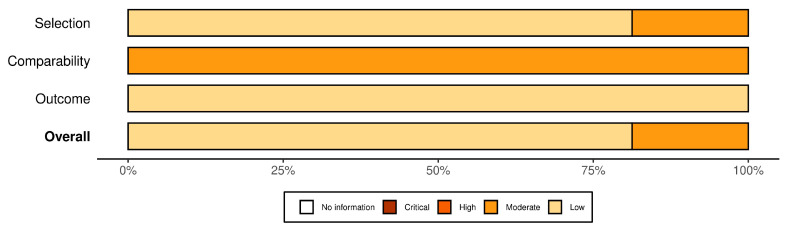
Summary plot of the quality assessment.

**Figure 3 healthcare-11-02066-f003:**
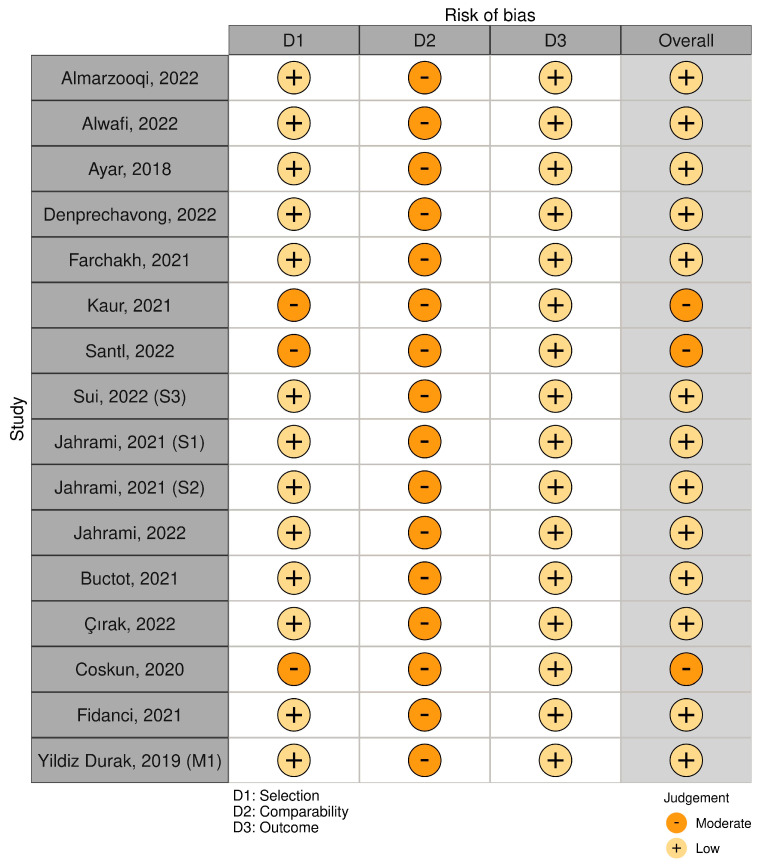
Traffic light plot of the quality assessment of the included studies [1,2,4,5,9,11,12,34,35,36,37,38,39,40,41].

**Figure 4 healthcare-11-02066-f004:**
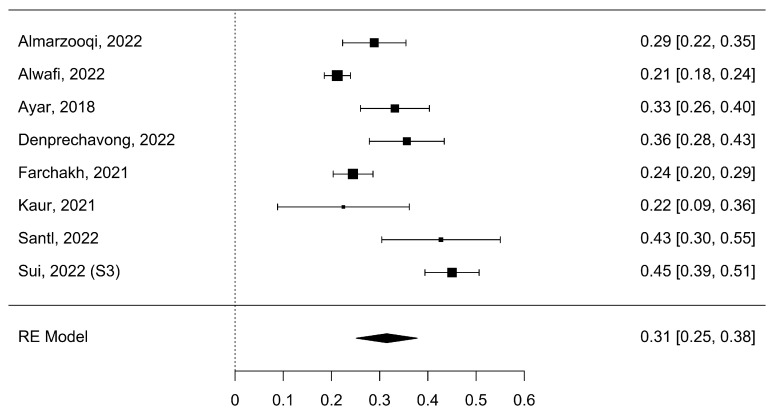
Meta-correlation of nomophobia symptoms with anxiety (NSA) [1,4,11,34,36,38,39,40].

**Table 1 healthcare-11-02066-t001:** Characteristics of the included studies.

SN	Ref.	Study	Country	Population Sample	Sample Size	Age (Years)	Insomnia	Smartphone Addiction	Anxiety	Insomnia’s Tool	Smartphone Addiction’s Tool	Anxiety’s Tool	Risk of Bias (NOS)
1	[11]	Almarzooqi et al., 2022	Saudi Arabia	General population	893	24	*r* = 0.253 *p* = 0.001	*r* = 0.158 *p* = 0.001	*r* = 0.281 *p* = 0.001	ISI	IAS	GAD-7	8 (low)
2	[34]	Alwafi et al., 2022	Saudi Arabia	General population	5191	24	NR	NR	*r* = 0.209 *p* = 0.001	NR	NR	RD	7 (low)
3	[1]	Ayar et al., 2018	Turkey	University students	755	21	NR	NR	*r* = 0.320 *p* = 0.001	NR	NR	SAAS	8 (low)
4	[9]	Buctot et al., 2021	Philippines	Adolescents	3374	15	NR	*r* = 0.116 *p* = 0.010	NR	NR	SAS-SV	NR	8 (low)
5	[5]	Çırak et al., 2022	Turkey	University students	451	20	NR	*r* = 0.579 *p* = 0.010	NR	NR	DAS	NR	8 (low)
6	[35]	Coskun et al., 2020	Turkey	General population	210	33	NR	*r* = NR *p* = 0.050	NR	NR	Scale of Social Media Addiction—Adult Form	NR	6 (moderate)
7	[36]	Denprechavong et al., 2022	Thailand	University students	638	20	NR	NR	*r* = 0.342 *p* = 0.001	NR	NR	HADS	8 (low)
8	[4]	Farchakh et al., 2021	Lebanon	General population	2260	28	*r* = 0.400 *p* = 0.001	NR	*r* = 0.240 *p* = 0.001	LIS-18	NR	HAM-A	8 (low)
9	[37]	Fidanci et al., 2021	Turkey	University students	386	22	NR	*r* = −0.053 *p* = 0.296	NR	NR	SAS-SV	NR	8 (low)
10	[2]	Jahrami et al., 2021 (S1)	Bahrain	General population	549	27	*r* = 0.630 *p* = 0.001	NR	NR	ISI	NR	NR	8 (low)
11	[2]	Jahrami et al., 2021 (S2)	Bahrain	General population	654	27	*r* = 0.600 *p* = 0.001	NR	NR	ISI	NR	NR	8 (low)
12	[12]	Jahrami et al., 2022	Bahrain	General population	549	27	*r* = 0.600 *p* = 0.001	NR	NR	ISI	NR	NR	8 (low)
13	[38]	Kaur et al., 2021	Pakistan	University students	209	21	NR	NR	*r* = 0.221 *p* = 0.001	NR	NR	SIAS	6 (moderate)
14	[39]	Santl et al., 2022	Croatia	Adolescents	257	22	NR	NR	*r* = 0.403 *p* = 0.010	NR	NR	DASS	6 (moderate)
15	[40]	Sui et al., 2022 (S3)	Canada	University students	1221	23	NR	NR	*r* = 0.422 *p* = 0.001	NR	NR	STAI	8 (low)
16	[41]	Yildiz Durak et al., 2019 (M1)	Turkey	Adolescents	612	13	NR	*r* = 0.819 *p* = 0.001	NR	NR	SAS	NR	7 (low)

NR, not recorded; GAD-7, Generalized Anxiety Disorder 7-item; SAAS, The Social Appearance Anxiety Scale; HADS, The Hospital Anxiety and Depression Scale; HAM-A, The Hamilton Anxiety Scale; DASS, The Depression, Anxiety, and Stress Scale; STAI, the State-Trait Anxiety Inventory; IAS, Internet Addiction Scale; SAS-SV, The Smartphone Addiction Scale—Short Version; DAS, The Digital Addiction Scale; ISI, Insomnia Severity Index; LIS-18, Lebanese Insomnia Scale.

**Table 2 healthcare-11-02066-t002:** Random effects meta-analysis models of nomophobia.

Analysis	Descriptive		Random-Effects Meta-Analysis	Prediction Intervals	Visual Results	Heterogeneity							Publication Bias	
	K	N	Pooled Results [95%CI]	PI [95%CI]	Forest Plot	τ^2^	τ	I^2^	H	df	Q	*p*	Egger’s Test	Rank Test
NSA	8	11,424	0.31 [0.25; 0.38]	[0.14; 0.49]	Figure 4	0.01	0.08	88.92%	9.02	7	72.93	<0.001	0.46	0.28
NSSA	6	5926	0.39 [0.04; 0.75]	[−0.54; 1.33]	Figure 5	0.20	0.44	99.32%	146.49	5	675.17	<0.001	0.86	0.72
NSI	5	4905	0.56 [0.38; 0.75]	[0.11; 1.01]	Figure 6	0.04	0.21	97.41%	38.58	4	141.37	<0.001	0.20	0.21

Note: NSA, nomophobia symptoms with anxiety; NSSA, nomophobia symptoms with smartphone addiction; NSI, nomophobia symptoms with insomnia; k = number of studies; n = number of participants; I^2^ statistic pointed to the percentage of variability through samples as a result of heterogeneity rather than chance; τ^2^ portrayed the degree of variability among the effects seen in different samples [between-sample variance]; H described the confidence intervals of heterogeneity.

## Data Availability

The data are available from the authors upon request.

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
