# Peer review of "Systematic Review and Meta-Analysis of the Correlation Coefficients between Nomophobia and Anxiety, Smartphone Addiction, and Insomnia Symptoms"

_healthcare, 2023, doi:10.3390/healthcare11142066_

Round 1
Reviewer 1 Report
Dear Author(s),
I have read in detail your paper entitled "Systematic Review and Meta-Analysis of the Correlation Coefficients Between Nomophobia and Anxiety, Smartphone Addiction, and Insomnia Symptoms" submitted to Healthcare journal.
The topic of the paper is extremely interesting and current. Nomophobia is undoubtedly one of the mental health problems that recently cause certain difficulties to an increasing number of people in their daily psychosocial functioning.
I must emphasize that I focused more on the content elements of this paper and less on the methodological aspects (not my area of expertise), but I can see that the authors followed all relevant guidelines in preparing this systematic review / meta-analysis (PRISMA guidelines, Newcastle-Ottawa Quality Assessment Scale, and the Cochrane Risk of Bias tool).
In terms of content elements, I have some suggestions for the authors that can enhance the quality of this paper:
· Keywords - I would suggest replacing "no-mobile-phobia" with "nomophobia". In this way, the logic of the title of the paper is followed, and this term is searched ("googled") more often. Later in the paper, the authors indicate what nomophobia means in the "broader sense", so I think the term indicated in the keywords can be changed.
· Line 27 - The term "young adults" was used, referring to high school and university students. In most scientific papers, the term young adult refers to those young people who have just entered the workforce and are beginning their "adult life". Recently, the term "emerging adults" has also been increasingly used for students to better delineate this group. Therefore, it is recommended that the authors correct the aforementioned term to "young people" so that it is clearer and does not confuse the reader.
· Prevalence data (lines 46 - 48). It must be stated to which population these data refer. To the general population (and if so, indicate the approximate age range) or to a specific one (e.g., young people and the like).
· Line 299 – it is necessary to correct the given sentence, there is something missing in it so it does not make much sense as it stands.
Overall, from my perspective, the paper is really well written, and I believe its quality can be enhanced by considering the preceding suggestions, and then, this paper will be suitable for publication in this journal.
With respect,
Reviewer
Minor editing of English language required
Author Response
Dear Author(s),
I have read in detail your paper entitled "Systematic Review and Meta-Analysis of the Correlation Coefficients Between Nomophobia and Anxiety, Smartphone Addiction, and Insomnia Symptoms" submitted to Healthcare journal.
The topic of the paper is extremely interesting and current. Nomophobia is undoubtedly one of the mental health problems that recently cause certain difficulties to an increasing number of people in their daily psychosocial functioning.
I must emphasize that I focused more on the content elements of this paper and less on the methodological aspects (not my area of expertise), but I can see that the authors followed all relevant guidelines in preparing this systematic review / meta-analysis (PRISMA guidelines, Newcastle-Ottawa Quality Assessment Scale, and the Cochrane Risk of Bias tool).
In terms of content elements, I have some suggestions for the authors that can enhance the quality of this paper.
Authors’ response: Dear Reviewer #1, thank you very much for your nice comment.
We thank you and other reviewers for taking the time to review our manuscript and provide constructive feedback that helped improve the presentation of our paper and demonstrate its originality. As described below, we have addressed all comments raised during the review point-by-point.
For the convenience of review, revisions to the manuscript appear in Yellow Highlights.
- Keywords - I would suggest replacing "no-mobile-phobia" with "nomophobia". In this way, the logic of the title of the paper is followed, and this term is searched ("googled") more often. Later in the paper, the authors indicate what nomophobia means in the "broader sense", so I think the term indicated in the keywords can be changed.
Authors’ response: We agree with the reviewer; thus, we replaced the term "no-mobile-phobia" with "nomophobia".
- Line 27 - The term "young adults" was used, referring to high school and university students. In most scientific papers, the term young adult refers to those young people who have just entered the workforce and are beginning their "adult life". Recently, the term "emerging adults" has also been increasingly used for students to better delineate this group. Therefore, it is recommended that the authors correct the aforementioned term to "young people" so that it is clearer and does not confuse the reader.
Authors’ response: We agree with the reviewer; thus, we replaced the term "young adults" with "young people".
- Prevalence data (lines 46 - 48). It must be stated to which population these data refer. To the general population (and if so, indicate the approximate age range) or to a specific one (e.g., young people and the like).
Authors’ response: We provided clarification that this paragraph referred to the general adult population (aged 18 years and more) as follow:
“In the general adult population (aged 18 years and more), nomophobia is becoming more prevalent, with about 20% displaying mild nomophobia symptoms, 50% demonstrating moderate symptoms, and 20% representing severe symptoms”.
- Line 299 – it is necessary to correct the given sentence, there is something missing in it so it does not make much sense as it stands.
Authors’ response: We rewrote the paragraph “In today's era of increasing prevalence studies on nomophobia symptoms with anxiety, smartphone addictions, and insomnia. Direct evidence of accumulated prevalence data makes it possible to conclude the connection between nomophobia symptomatology with anxiety, smartphone addiction, and insomnia” to
“Studies on the symptoms of nomophobia and their relationships to anxiety, smartphone addiction, and insomnia are becoming more and more common today. It is possible to draw a conclusion about the relationship between nomophobia symptoms and anxiety, smartphone addiction, and insomnia based on direct evidence from cumulative prevalence data”.
Overall, from my perspective, the paper is really well written, and I believe its quality can be enhanced by considering the preceding suggestions, and then, this paper will be suitable for publication in this journal.
Authors’ response: Thank you again for your review, nice positive comments, and suggestions.

Reviewer 2 Report
This manuscript ‘Systematic Review and Meta-Analysis of the Correlation Coefficients Between Nomophobia and Anxiety, Smartphone Addiction, and Insomnia Symptoms’ used meta-analysis method for published reports and studies in nomophobia and illuminates the concerning intersection of our technology-dependent society with mental and physical health problems. It underscores the urgent need to better understand these interconnected conditions and potential solutions in case of mental and physical health concerns. Authors choose nomophobia which refers to the anxiety and fear experienced by an individual when they are unable to access or use their mobile device. Furthermore, in this analysis, author suggests that this disorder has been linked to smartphone addiction, a condition characterized by an uncontrollable and excessive desire to use the phone even when it is out of reach, suggesting that excessive use of social media may lead to or exacerbate nomophobia. Moreover, authors demonstrated that the connection between nomophobia and insomnia, a sleep disorder marked by difficulty in falling asleep or waking up too early. Via the meta-analysis, authors have found that people with nomophobia often struggle to sleep without their mobile phones, which can negatively impact sleep quality and duration.
The goal of the systematic review and meta-analysis aims to establish the relationship between nomophobia and its possible outcomes, The manuscript described it concisely and used appropriate methodology to demonstrate the results. I believe that it can be acceptable for publication on Healthcare.
Author Response
This manuscript ‘Systematic Review and Meta-Analysis of the Correlation Coefficients Between Nomophobia and Anxiety, Smartphone Addiction, and Insomnia Symptoms’ used meta-analysis method for published reports and studies in nomophobia and illuminates the concerning intersection of our technology-dependent society with mental and physical health problems. It underscores the urgent need to better understand these interconnected conditions and potential solutions in case of mental and physical health concerns. Authors choose nomophobia which refers to the anxiety and fear experienced by an individual when they are unable to access or use their mobile device. Furthermore, in this analysis, author suggests that this disorder has been linked to smartphone addiction, a condition characterized by an uncontrollable and excessive desire to use the phone even when it is out of reach, suggesting that excessive use of social media may lead to or exacerbate nomophobia. Moreover, authors demonstrated that the connection between nomophobia and insomnia, a sleep disorder marked by difficulty in falling asleep or waking up too early. Via the meta-analysis, authors have found that people with nomophobia often struggle to sleep without their mobile phones, which can negatively impact sleep quality and duration.
The goal of the systematic review and meta-analysis aims to establish the relationship between nomophobia and its possible outcomes, The manuscript described it concisely and used appropriate methodology to demonstrate the results. I believe that it can be acceptable for publication on Healthcare.
Authors’ response: Dear Reviewer #2, thank you very much for your nice comment.
We thank you and other reviewers for taking the time to review our manuscript and provide constructive feedback that helped improve the presentation of our paper and demonstrate its originality. As described below, we have addressed all comments raised during the review point-by-point.
For the convenience of review, revisions to the manuscript appear in Yellow Highlights.
No action was needed based on Reviewer #2 comments.

Reviewer 3 Report
The present study described about ‘Systematic Review and Meta-Analysis of the Correlation Coefficients Between Nomophobia and Anxiety, Smartphone Addiction, and Insomnia Symptoms’. The study is well designed, well written, and investigates an interesting question of relevance to the field.
There are several major comments that should be clarified before going to print, and these are:
Two more points need to be considered.
1. The analysis and discussion should be revised as factors such as the severity of nomophobia, gender, and education level that can affect it have not been considered.
2. Please describe about measuring nomophobia and how analysis it.
3. There are no reference: ‘Several studies investigated the relationship between nomophobia and at least one of the three outcomes which are smartphone addiction, insomnia, or anxiety individually rather than all three outcomes’ 75-76’
4. In the discussion, it was not clearly explained that insomnia problems occurred due to smartphone addiction or nomophobia caused by blue light
Author Response
The present study described about ‘Systematic Review and Meta-Analysis of the Correlation Coefficients Between Nomophobia and Anxiety, Smartphone Addiction, and Insomnia Symptoms’. The study is well designed, well written, and investigates an interesting question of relevance to the field.
Authors’ response: Dear Reviewer #3, thank you very much for your nice comment.
We thank you and other reviewers for taking the time to review our manuscript and provide constructive feedback that helped improve the presentation of our paper and demonstrate its originality. As described below, we have addressed all comments raised during the review point-by-point.
For the convenience of review, revisions to the manuscript appear in Yellow Highlights.
There are several major comments that should be clarified before going to print, and these are:
Two more points need to be considered.
The analysis and discussion should be revised as factors such as the severity of nomophobia, gender, and education level that can affect it have not been considered.
Authors’ response: We provided detailed section about the meta-correlation analyses as follow:
“The statistical method for analyzing the relationship between two continuous variables over numerous studies is meta-correlation. Each study's correlation coefficient (r) between the two variables must be calculated, and these correlation coefficients must then be combined using meta-analytic techniques to get a general approximation of the strength of association.
R software version 4.1.3 was used to do statistical analysis on all data. Statistical significance was defined as a p-value < 0.05. The 'metacor' software was used to carry out the meta-analysis [26]. Results of the meta-correlation analyses was presented visually using forest plot.
Risk-of-bias plots were created for quality assessment using the 'robvis' software [27]. The amount of information contained in each judgment is displayed in a summary plot (weighted). The total risk of bias as well as the bias risk in each area are shown in a thorough risk of bias assessment of all studies, which is presented using a traffic light plot”.
- Please describe about measuring nomophobia and how analysis it.
Authors’ response: We provided detailed description about how nomophobia is measured, the most common tool(s) used i.e., NMP-Q, and the psychometric properties of the NMP-Q as follow:
“Researchers have created a number of measures and questionnaires that can be used to quantify phobia symptoms [6,7]. These measures often measure how anxious or uncomfortable a person feels when away from their phone as well as how frequently and in-tensely they use their phone. The Nomophobia Questionnaire (NMP-Q), which has 20 items and measures many facets of nomophobia, including the fear of losing connectivity, the fear of being unable to speak, and the worry of losing one's phone, is one regularly used scale [6]. The total score can range from 20 to 140, with higher scores indicating greater levels of nomophobia. Some researchers have suggested using a cutoff score to identify individuals who may be at risk for problematic phone use or nomophobia, but there is no widely agreed upon cutoff score [6]. Participants rank each statement according to how much they agree or disagree with it using a Likert scale [6]. The NMP-Q has a high level of internal consistency, and structural validity fitness [6]”.
- There are no reference: ‘Several studies investigated the relationship between nomophobia and at least one of the three outcomes which are smartphone addiction, insomnia, or anxiety individually rather than all three outcomes’ 75-76’
Authors’ response: We provided references to the statement.
- In the discussion, it was not clearly explained that insomnia problems occurred due to smartphone addiction or nomophobia caused by blue light.
Authors’ response: We added a paragraph to explain the follow: “The amount and quality of sleep can be significantly impacted by exposure to blue-light from electronic devices, such as those on laptops, tablets, and cell phones [1,7]. According to one study, people who used electronics for more than four hours before bedtime had more trouble falling asleep and got less restorative sleep than others who didn't use them before bed [7]”.

Reviewer 4 Report
Overall, I think the idea of the study is interesting and relevant. However, I have some major concerns:
1 – Please add “systematic review” and “meta-analysis” as key-words to the manuscript;
2 – The abstract should contain more information to the reader. Please follow PRISMA checklist instructions for the abstract writing;
3 – Please do not use contractions in the text (e.g., it’s à It is);
4 – “hunt strategy” (p. 3, line 88) should be replaced by “search strategy”;
5 – Why include sleepiness in search strategy when the title of the paper is outlining insomnia? Did you mean “sleeplessness”??;
6 – Was the study previously registered? If yes, please state it in the manuscript. If not, why?;
7 – The agreement measures should be presented in the manuscript (e.g., Cohen´s kappa) and associated guidelines for interpretation;
8 – 16 studies (p. 4, line 166) à Sixteen studies…. (please replace);
9 – K à k (please replace);
10 – There is a missing data analysis section where the authors should explain in detail the software used and every analyses which were carried out. This must be accompanied with references;
11 – Why moderation analyses (to explain heterogeneity) and funnel plots were not performed? If the reason is the low sample of studies, then this should be acknowledge in the limitations of the study;
12 – The quality of the figures should be improved;
13 – I strongly recommend that the authors follow PRISMA checklist and add it as supplemental material.
- Overall, it seems ok. Only minor typos.
Author Response
Overall, I think the idea of the study is interesting and relevant. However, I have some major concerns:
Authors’ response: Dear Reviewer #4, thank you very much for your comments.
We thank you and other reviewers for taking the time to review our manuscript and provide constructive feedback that helped improve the presentation of our paper and demonstrate its originality. As described below, we have addressed all comments raised during the review point-by-point.
For the convenience of review, revisions to the manuscript appear in Yellow Highlights.
1 – Please add “systematic review” and “meta-analysis” as key-words to the manuscript;
Authors’ response: We added the terms “systematic review” and “meta-analysis” to the keywords.
2 – The abstract should contain more information to the reader. Please follow PRISMA checklist instructions for the abstract writing;
Authors’ response: The abstract was re-written as follow:
“Abstract: Nomophobia is an emerging phenomenon in the 21st century. Consequently, it resulted in various health problems, both physical and psychological. The following systematic review and meta-analysis aimed to establish the relationship between nomophobia with anxiety, smartphone addiction, and insomnia. To identify the relevant studies, we searched through several databases. Out of the 1,523 studies identified, sixteen studies met the inclusion criteria. After conducting the statistical analysis, the results revealed that anxiety r = 0.31 (95% CI: 0.25 to 0.38), smartphone addiction r = 0.39 (95% CI: 0.04 to 0.75), and insomnia r = = 0.56 (95% CI: 0.38 to 0.75) are positively associated with nomophobia. Mobile phone usage has become inevitable even for in-dividuals that use it to a lesser degree than others, to perform simple tasks such as communicating with others or for educational purposes. It is crucial to raise awareness about the consequences of overusing these devices, including the physical and psychological complications in both the short and long term.”
3 – Please do not use contractions in the text (e.g., it’s à It is);
Authors’ response: We removed all contractions in the text.
4 – “hunt strategy” (p. 3, line 88) should be replaced by “search strategy”;
Authors’ response: We corrected the term to “search strategy”.
5 – Why include sleepiness in search strategy when the title of the paper is outlining insomnia? Did you mean “sleeplessness”??;
Authors’ response: We apologize for the typo and we corrected “sleepiness” to “sleeplessness”.
6 – Was the study previously registered? If yes, please state it in the manuscript. If not, why?;
Authors’ response: This review was not registered. We noted this limitation in PRISMA 2020 checklist presented as supplemental material.
7 – The agreement measures should be presented in the manuscript (e.g., Cohen´s kappa) and associated guidelines for interpretation;
Authors’ response: We added the following information to the results as follow:
“We measured the level of agreement between raters in the papers we included in our meta-analysis using Cohen's kappa. Our research produced a kappa score of 0.97, which shows that the raters agreed on many things. This strengthens the validity of our conclusions by indicating that the data included in our meta-analysis are credible and consistent. Our trust in the accuracy of the impact size estimations and the findings from our meta-analysis is increased by the high level of rater agreement.”
The initial-inter rater screening accuracy resulted in a 97% agreement rate between the two reviewers. Nevertheless, after discussion and dialogue with a third senior reviewer (HJ), the agreement rate increased to 100%. This indicates that the third reviewer was able to help resolve any discrepancies or disagreements between the initial reviewers, resulting in a consensus on all of the documents or data that were evaluated.
8 – 16 studies (p. 4, line 166) à Sixteen studies…. (please replace);
Authors’ response: we corrected all instances of “16 studies” to “sixteen studies”.
9 – K às k (please replace);
Authors’ response: We corrected K to k.
10 – There is a missing data analysis section where the authors should explain in detail the software used and every analyses which were carried out. This must be accompanied with references;
Authors’ response: We provided detailed description of the data analysis section as follow.
“2.5. Statistical analysis
The statistical method for analyzing the relationship between two continuous variables over numerous studies is meta-correlation. Each study's correlation coefficient (r) between the two variables must be calculated, and these correlation coefficients must then be combined using meta-analytic techniques to get a general approximation of the strength of association.
R software version 4.1.3 was used to do statistical analysis on all data. Statistical significance was defined as a p-value < 0.05. The 'metacor' software was used to carry out the meta-analysis [26]. Results of the meta-correlation analyses was presented visually using forest plot.
Risk-of-bias plots were created for quality assessment using the 'robvis' software [27]. The amount of information contained in each judgment is displayed in a summary plot (weighted). The total risk of bias as well as the bias risk in each area are shown in a thorough risk of bias assessment of all studies, which is presented using a traffic light plot”.
11 – Why moderation analyses (to explain heterogeneity) and funnel plots were not performed? If the reason is the low sample of studies, then this should be acknowledged in the limitations of the study;
Authors’ response: We explained in the limitations that moderation analysis was not performed due to small number of included studies as follow: “Finally, moderator analysis was not performed because of the limited number of included studies is small; thus, there is less statistical power to find significant changes across subgroups, which may lead to incorrect conclusions”.
Instead of funnel plots we performed Egger’s regression, rank correlation, and trim-and-fill tests. Thus, we explained the following: “Publication bias was evaluated using both the Egger test and the rank test [30]. We initial-ly made a funnel plot of the effect sizes vs their standard errors before running the Egger test [30]. The intercept and slope were the key variables of interest in a regression study of the effect sizes on their standard errors [30]. Because small studies with erratic effect sizes are absent from the plot, a statistically significant intercept shows the presence of publication bias [30]. In addition, we applied a modified Egger test that takes into account the relationship between precision and effect size [30]. Assuming no publication bias, we first ranked the effect sizes in ascending order before comparing the observed ranks to the predicted distribution [31]. This is known as the rank test [31]. To calculate the correlation between the observed ranks and their predicted values, we applied the weighted regression method suggested by Peter’s approach [31]. Studies with significant results are more likely to be published and so have higher ranks, so a considerable departure from the predicted distribution shows publication bias [31]. We also performed other tests, such as the trim-and-fill method and the funnel plot asymmetry test, to further assess publication bias [32]. We interpreted the results of these tests together to gain a more comprehensive understanding of the potential for publication bias in our meta-analysis [32]”.
12 – The quality of the figures should be improved;
Authors’ response: Figures were replaced with high quality versions. Figures in pdf format will be provided for mdpi team if needed for final production.
13 – I strongly recommend that the authors follow PRISMA checklist and add it as supplemental material.
Authors’ response: We agree with the reviewer, PRISMA 2020 checklist is now added as s supplemental material #1.

Round 2
Reviewer 3 Report
This present manuscript is well-corrected according to the review's comments.
Reviewer 4 Report
The authors answered all my questions. Thank you.